

# A generalized fuzzy clustering framework for incomplete data by integrating feature weighted and kernel learning

Ying Yang[1], Haoyu Chen[2] and Haoshen Wu[3]

[1] College of Information and Intelligence, Hunan Agricultural University, Changsha, China
[2] New Energy College, Xi'an Shiyou University, Xi'an, China
[3] College of Management, Guangdong University of Technology, Guangzhou, China

## ABSTRACT

Missing data presents a challenge to clustering algorithms, as traditional methods tend to pad incomplete data first before clustering. To combine the two processes of padding and clustering and improve the clustering accuracy, a generalized fuzzy clustering framework is proposed based on optimal completion strategy (OCS) and nearest prototype strategy (NPS) with four improved algorithms developed. Feature weights are introduced to reduce outliers' influence on the cluster centers, and kernel functions are used to solve the linear indistinguishability problem. The proposed algorithms are evaluated regarding correct clustering rate, iteration number, and external evaluation indexes with nine datasets from the UCI (University of California, Irvine) Machine Learning Repository. The results of the experiment indicate that the clustering accuracy of the feature weighted kernel fuzzy C-means algorithm with NPS (NPS-WKFCM) and feature weighted kernel fuzzy C-means algorithm with OCS (OCS-WKFCM) under varying missing rates is superior to that of seven conventional algorithms. Experiments demonstrate that the enhanced algorithm proposed for clustering incomplete data is superior.

## INTRODUCTION

All areas are flooded with massive amounts of data with complex trends. Clustering analysis (*Sinaga & Yang, 2020*) is an unsupervised learning technique, which can autonomously classify data without *a priori* knowledge. Additionally, it is one of the effective tools to fully exploit the value present in the data. The traditional hard clustering approach considers that data objects can be grouped entirely into a certain category. However, in real life, there are no clear boundaries for many things. *Bezdek (1981)* introduced the fuzzy set theory (*Zadeh, Klir & Yuan, 1996*) into the clustering algorithm and proposed the FCM algorithm. The algorithm represents the relationship between data and clusters with an affiliation value of 0-1, which is more suitable for practical clustering problems. Nevertheless, the FCM algorithm cannot directly cluster incomplete datasets. However, missing datasets are more prevalent in real-world fields such as industry,

Corresponding author
Ying Yang,
1254809709@stu.hunau.edu.cn

medicine, business and scientific research (*Ma et al., 2021*; *Babaee Khobdeh, Yamaghani & Khodaparast Sareshkeh, 2021*; *Ma et al., 2020a*). Nearly 45% of the datasets in the UCI Machine Learning Repository are missing relevant data. Not only do missing data result in the loss of a substantial quantity of valuable information, but they also present difficulties for cluster analysis. Therefore, it is of great practical importance to investigate fuzzy clustering algorithms for incomplete data.

Numerous researchers have proposed enhanced algorithms to address the issue of FCM clustering of insufficient data. The most classic of these are the four improved fuzzy clustering algorithms for incomplete data proposed by *Hathaway & Bezdek (2001)*. Based on the Whole Data Strategy (WDS), Partial Distance Strategy (PDS), Optimized Complete Strategy (OCS), and Nearest Prototype Strategy (NPS), four algorithms are preposed. The WDS-FCM algorithm is a rounding method that discards missing values. The PDS-FCM algorithm improves the formulation of the FCM clustering algorithm by introducing the local distance introduced by *Dixon (1979)* without considering missing values in the calculation to fulfill incomplete data clustering. The OCS-FCM algorithm continuously interpolates absent values as updateable variables. In addition, the NPS-FCM algorithm replaces absent values with attribute values corresponding to clustering centers closest to the incomplete data. The four algorithms provide effective ideas for the interpolation of incomplete data.

Among the four strategies, the OCS and the NPS are more widely adopted and continuously improved by researchers. *Li et al. (2017)* proposed an interval kernel fuzzy C-means clustering method for incomplete data by converting the incomplete data set into an interval data set and introducing the NPS-based kernel method. *Najib et al. (2020)* modified the NPS-FCM algorithm based on the continuous mechanism so that it can be used to aggregate incomplete data streams with high error rates. *Yenny et al. (2021)* presented a cluster intelligence-based framework for clustering incomplete data using a swarm intelligence algorithm to determine cluster centers and hyperparameters. *Shi & Wang (2022)* proposed a clustering algorithm based on the relationship between attributes, which combines support vector machines with the four clustering strategies mentioned above.

In addition, another solution for clustering incomplete data is to first interpose the missing values by evaluation and then cluster the completed dataset. Due to the few parameters and straightforward principle of the K-nearest neighbor (KNN) algorithm, it is gaining popularity for interpolating incomplete data (*Xiuqin et al., 2023*). *Ma et al. (2020b)* constructed models based on genetic programming (GP) using other available features to predict missing values for incomplete features and used a weighted K-nearest neighbors (KNN) for the selection of instances. *Al-Helali et al. (2021)* proposed a new incomplete pattern belief classification (PBC) method by combining multiple estimates of nearest neighbors (KNN). *Qi, Guo & Wang (2021)* proposed a reliable k-nearest neighbor method (RKNN) for incomplete interval-valued data (IIDD). *Gao, Yuan & Tang (2022)* improved the K-nearest neighbor (KNN) algorithm with the help of Patial Distance Strategy (PDS) by applying it to incomplete datasets. *Baligh et al. (2021)* presented a novel

genetic programming and weighted KNN-based interpolation method for incomplete data regression.

Based on the idea of expectation-maximization (EM), the corresponding incomplete data processing and clustering methods are proposed. *Ruggieri et al. (2020)* improved the EM algorithm based on Bayesian Networks (BN) to enable the algorithm to estimate incomplete data. *Maghsoodi et al. (2023)* estimated incomplete data using Expectation Maximization (EM) algorithm and extracted core criteria using Recursive Feature Elimination (RFE) with Least Square Support Vector (LS-SVM). *Wang et al. (2020)* retained the information hidden in the missing data based on the expectation maximization (EM) algorithm and Bayesian network (BN) approach for attribution of missing data, *i.e.,* EM-BN approach.

With intensive research and development, neural networks are also used to process incomplete data. *Sovilj et al. (2016)* introduced a multiple valuation algorithm for incomplete data based on the Gaussian mixture model and extreme learning machine. *Truong et al. (2020)* proposed an effective deep feedforward neural networks (DFNN) method for damage identification of truss structures based on noisy incomplete modal data. *Dai, Bu & Long (2023)* proposed multiple imputation (MI) methods based on neural network Gaussian process (NNGP) for estimation of incomplete data. *Xu et al. (2022)* proposed a practical method based on Physical Information Neural Network (PINN) to combine known data with physical principles to reconstruct the flow field with imperfect data.

After filling the incomplete dataset with various interpolation methods, the second step is to perform clustering. Several experts have improved the clustering algorithm from the perspective of dataset attributes (*Tran et al., 2018*). *Li & Wei (2020)* proposed a feature-weighted K-means clustering method based on two distance measures, dynamic time warping (DTW) and shape-based distance (SDB). *Paul & Das (2020)* proposed a Gibbs sampling-based algorithm for the Dirichlet process mixture model that determines the number of clusters and also incorporates near-optimal feature weighting. *Ghodratnama & Abrishami (2021)* clusters the data iteratively by using a supervised C-means method and weights the features in each cluster using a local feature weighting method. *Yang & Benjamin (2021)* proposed a feature-weighted reduced PCM algorithm (FW-R-PCM) that can be used to identify important features by calculating feature weights.

The three interpolation methods mentioned above all present different disadvantages. KNN filling-based clustering methods can achieve better results only in large-scale sparse data with few values of missing attributes. The EM-based clustering methods often fail to obtain the desired filling effect when there is a large amount of missing data, or a certain large class of values is missing. The neural network-based clustering methods require a large amount of model training to estimate the missing values of individual missing instances, which greatly increases the computational cost. Although the clustering improvement methods that introduce feature weighting and kernel functions (*Vo, Nguyen & Vo, 2016*) are effective, methods that split the interpolation and clustering ultimately lead to a secondary reduction of computational accuracy.

So far, it is still an open issue how to effectively solve the clustering task for incomplete data. To enhance the performance of incomplete data clustering tasks, we therefore propose

a generalized fuzzy clustering framework integrating feature weights and kernel learning. Currently, a number of experiments conducted on public data sets demonstrate the efficacy and superiority of the proposed method. The following are the primary contributions of this work:

On the basis of OCS and NPS in literature (*Hathaway & Bezdek, 2001*), we unify imputation, feature learning, and clustering as one optimization objective, and propose OCS-WFCM and NPS-WFCM, respectively.

In order to better adapt to incomplete data clustering in complex cases (*e.g.*, non-linear data), we further propose kernel-based OCS-WKFCM and NPS-WKFCM methods.

An alternate optimization method is used to solve the objective functions of the above methods, and the optimal solutions are obtained by iterative updating of variables.

The research is structured as follows. In Section 'Analysis of incomplete data clustering algorithm', the FCM algorithm theory and four strategies for incomplete data are analyzed in detail. In Section 'Feature weighted kernel function FCM of incomplete data', four improved algorithms based on the established framework are introduced. In Section 'Experimental evaluation', comparing the four algorithms proposed in this research to other fragmentary data clustering algorithms verifies the framework's efficacy. Finally, the summaries and optimizations are given in Section 'Conclusion'.

# ANALYSIS OF INCOMPLETE DATA CLUSTERING ALGORITHM

## Fuzzy C-means algorithm

FCM algorithm's fundamental concept is to minimize objective function to solve clustering center and membership matrix. The primary implementation process is to establish the objective function formula based on the data sample's proximity to the clustering centroid. Iteratively updating the membership moment clustering center matrix, the algorithm determines the objective function's extreme point. Finally, the category of the data sample is determined according to the size of the membership value (*Askari, 2021*).

Let $U_{(c \times n)}$ represent the membership matrix, and $V$ represent the cluster center matrix. Suppose a dataset $X = \{x_1, x_2, \ldots, x_n\}$ exists in $s$ dimensions and $n$ samples. The dataset can be represented as $x_k = [x_{1k}, x_{2k}, \ldots, x_{sk}]^T$, and the samples can be defined as $x_{ik}$. The number of sample clusters in the dataset is set to $c$, the membership value of data $x_j$ to category $i$ is expressed as $u_{ij} \in U_{(c \times n)}$. The sample $x_k$ is characterized by different affiliation values for different clusters, and the sum of c categories' membership values is 1. That is, $u_{ij}$ is shown in the constraint formula (Eq. 2.1).

$$\sum_{i=1}^{c} u_k = 1, \ k = 1, 2, \ldots, n \text{ and } \sum_{i=1}^{c} \mu_{ik} = 1$$
$$0 < u_{ik} < 1, \ \forall i = 1, 2, \ldots, c \tag{2.1}$$

The objective function formula established by FCM is shown in (Eq. 2.2).

$$\min \sum_{i=1}^{c} \sum_{k=1}^{n} u_{ik}^{m} \|x_k - v_i\|_2^2 \tag{2.2}$$

Where, $m$ is the fuzzy weighting coefficient, $\|\cdot\|_2$ is normal form, cluster center $V = \{v_1, v_2, \ldots, v_c\}$, and the membership matrix $U_{(c \times n)}$, $J(U, V)$ equals the sum of the sample cluster squares and the cluster center.

Lagrange multiplier method is used to solve the multivariate function's extreme value, which is used to solve membership function matrix and clustering center function matrix of FCM algorithm. The membership updating formula is shown in (Eq. 2.3).

$$u_{ik} = \left[ \sum_{t=1}^{c} \left( \frac{\|x_k - v_i\|_2^2}{\|x_k - v_t\|_2^2} \right)^{\frac{1}{m-1}} \right]^{-1}, i = 1, 2, \ldots, c; k = 1, 2, \ldots, n \tag{2.3}$$

The cluster center update formula is shown in (Eq. 2.4).

$$v_i = \frac{\sum_{k=1}^{n} u_{ik}^m x_k}{\sum_{k=1}^{n} u_{ik}^m}, i = 1, 2, \ldots, c \tag{2.4}$$

## Improved FCM algorithm for incomplete data

*Hathaway & Bezdek (2001)* have proposed four classical fuzzy clustering algorithms for incomplete data. The data set information is described as follows:

$\bar{X} = \{\bar{x}_1, \bar{x}_2, \ldots, \bar{x}_n\}$ is an incomplete data set, the single sample data in the data set is expressed as $\bar{x}_i = [\bar{x}_{1i}, \bar{x}_{2i}, \ldots, \bar{x}_{si}]^T (1 \leq k \leq n)$, and the number of attribute values is $s$.

$\bar{X}$ will be divided into two types of data sample sets: complete data set $X_W = \{x_k \in X | x_k \text{ is the complete data sample}\}$ and incomplete data sample set $\tilde{X}_N = \{\tilde{x}_k \in \tilde{X} | \tilde{x}_k \text{ is an incomplete data sample}\}$. The attribute information set is divided into two categories: $1 \leq j \leq s$, $1 \leq k \leq n$, complete data set $X_P = \{x_{jk} | x_{jk} \text{is the complete attribute}\}$, and missing attribute set $X_M = \{x_{jk} = |x_{jk} \text{is the missing attribute}\}$.

### FCM algorithm with whole data strategy

In the WDS-FCM algorithm, a simple method is used to directly discard the samples with missing attributes. Then, the data samples in the sample set $X_P = \{x_{jk} | x_{jk} \text{is the complete attribute}\}$ are directly clustered by FCM.

The dealing strategy of WDS-FCM algorithm will cause data samples with missing attributes to discard other complete attributes. This can result in a large amount of wasted data information. When the missing rate in the dataset is low, it has little effect on the overall dataset. With the remaining complete sample for fuzzy clustering, the calculated clustering center is not much different from the original data clustering center. Due to the absence of a large number of attributes, the clustering accuracy will be significantly impacted by an increase in the missing rate. Therefore, *Hathaway & Bezdek (2001)* suggest that WDS-FCM algorithm is more suitable for clustering analysis of datasets, as the proportion of missing attribute information in incomplete datasets is less than 0.25.

The WDS-FCM algorithm proceeds as follows:

(1) Split data $X$: the incomplete data set is separated into two sections: the complete part $X_P$, the missing part $X_M$, and $X = X_P \cup X_M$. In the experiment, $X_P$ instead of $X$, $X_M$ in FCM algorithm does not participate in the calculation.

(2) Initialization: iterative convergence threshold $\varepsilon$, fuzzy parameter $m$, cluster number $c (2 \leq c \leq \sqrt{n})$, maximum number of iterations $G$, initial membership matrix $U^{(0)}$.

(3) Updating the cluster center: when the algorithm performs $L$ ($L = 1, 2, \ldots$) iterations, cluster center $V^{(l)}$ is updated according to $U^{(l-1)}$ and the cluster center calculation formula (Eq. 2.4).

(4) Calculation of membership matrix: according to $V^{(l)}$ and (Eq. 2.3), solve membership matrix $U^{(l)}$.

(5) Iteration termination: when the iteration count approaches $L = G$, or $\forall i, k$, $max \left| u_{ik}^{(l)} - u_{ik}^{(l-1)} \right| < \varepsilon$, WDS-FCM algorithm iteration stops, the algorithm ends, the output membership $U$ and cluster center $V$; or else $L = L + 1$, return (3) to continue.

### FCM algorithm with partial distance strategy

On the basis of WDS-FCM, PDS-FCM in terms of attributes, the attributes participate in calculating local distances as long as they exist. When the attribute is missing, the complete attribute participation is converted. The distance between missing data sample $x_k$ and cluster center $v_i$ is determined according to attribute ratio.

$$D_{ik} = \frac{s}{\sum_{j=1}^{s} I_{jk}} \sum_{j=1}^{s} \left( x_{jk} - v_{ji} \right)^2 I_{jk} \tag{2.12}$$

Among them,

$$I_{jk} = \begin{cases} 0, & if \ x_{jk} \in \tilde{X}_M \\ 1, & if \ x_{jk} \in \tilde{X}_P \end{cases}, 1 \leq j \leq s, 1 \leq k \leq n \tag{2.13}$$

The clustering center at the extremum point is as follows.

$$v_{ji} = \frac{\sum_{k=1}^{n} \mu_{ik}^m I_{jk} x_{jk}}{\sum_{k=1}^{n} \mu_{ik}^m I_{jk}}, 1 \leq j \leq s, 1 \leq i \leq c \tag{2.14}$$

The membership formula is shown as (Eq. 2.15).

$$u_{ik} = \left[ \sum_{t=1}^{c} \left( \frac{\|x_k - v_i\|_2^2}{\|x_k - v_t\|_2^2} \right)^{\frac{1}{m-1}} \right]^{-1}, i = 1, 2, \ldots, c; k = 1, 2, \ldots, n \tag{2.15}$$

The PDS-FCM algorithm proceeds as follows:

(1) Initialization: iterative convergence threshold $\varepsilon$, fuzzy parameter $m$, cluster number $c \left( 2 \leq c \leq \sqrt{n} \right)$, maximum number of iterations $G$, initial membership matrix $U^{(0)}$.

(2) Updating the cluster center: when the algorithm performs $L$ ($L = 1, 2, \ldots$) iterations, the cluster center $V^{(l)}$ is updated according to $U^{(l-1)}$ and (Eq. 2.14).

(3) Calculating the membership matrix: according to $V^{()l()}$ and (Eq. 2.15), solving membership matrix $U^{(l)}$.

(4) Iteration termination: when the iteration count approaches $L = G$, or $\forall i, k$, $max \left| u_{ik}^{(l)} - u_{ik}^{(l-1)} \right| < \varepsilon$, PDS-FCM algorithm iteration stops, the algorithm ends, the output membership $U$ and cluster center $V$; or else $L = L + 1$, return (3) to continue.

### FCM algorithm with optimal completion strategy

The OCS-FCM algorithm assigns the lacking attributes as variables and incorporates variables into the objective function calculation of the FCM algorithm. Iterative clustering is performed with variables instead of missing attributes.

The variable membership $U$ and the cluster center $V$ are iteratively updated in the clustering iteration process to find the optimal value. The objective function formula established by OCS-FCM is (Eq. 2.16).

$$\min \sum_{i=1}^{c} \sum_{k=1}^{n} u_{ik}^{m} \|\tilde{x}_k - v_i\|_2^2. \tag{2.16}$$

Using the Lagrange multiplier method to locate the extremum of objective function (Eq. 2.16), the missing attribute update formula (Eq. 2.17) is obtained.

$$x_{jk} = \frac{\sum_{i=1}^{c} u_{ik}^{m} v_{ji}}{\sum_{i=1}^{c} u_{ik}^{m}}. \tag{2.17}$$

The main steps of OCS-FCM algorithm are:

(1) Initialization: Set the fuzzy parameter m, number of clusters $c \left(2 \leq c \leq \sqrt{n}\right)$, utmost allowed iterations $G$, iterative convergence threshold $\varepsilon$, the missing attribute matrix $\tilde{X}_M^{(0)}$, and the membership matrix $U^{(0)}$ combined with the constraint conditions.

(2) Updating the cluster center matrix: when the algorithm performs $L$ $(L = 1, 2, \ldots)$ iterations, the cluster center $V^{(l)}$ is updated according to $U^{(l-1)}$ and (Eq. 2.3).

(3) Calculate the membership matrix: according to $V^{(l)}$, and (Eq. 2.4) solving membership matrix $U^{(l)}$.

(4) Update the missing value: calculate the missing value $\tilde{X}_M^{(l)}$ according to the membership partition matrix $U^{(l)}$ and cluster center matrix $V^{(l)}$ and (Eq. 2.17).

(5) Iteration termination: when the iteration count approaches $L = G$, or $\forall i, k$, $max \left|u_{ik}^{(l)} - u_{ik}^{(l-1)}\right| < \varepsilon$, OCS-FCM algorithm iteration stops, and the program outputs membership $U$ and cluster center $V$; or else $L = L + 1$, return (3) to continue.

### FCM algorithm with the nearest prototype strategy

The NPS-FCM algorithm is an estimation method. In the NPS-FCM algorithm, the missing data attributes in the NPS-FCM algorithm participate in clustering with the nearest neighbor center instead. The missing data no longer remain constant after pre-population. During the iterative process, the corresponding attribute values of the clustering centers are continuously followed and adjusted. The filling method for missing attributes is as follows.

$$x_{jk}^{(l)} = v_{ji}, D_{ik} = \min\{D_{1k}, D_{2k}, \ldots, D_{ck}\}. \tag{2.18}$$

The NPS-FCM algorithm is based on the OCS-FCM algorithm. In the process of iteration, the missing data attribute is replaced by (Eq. 2.18), and then the clustering analysis is performed according to the implementation steps of the OCS-FCM algorithm.

The main steps of NPS-FCM algorithm are:

(1) Initialization: Set the fuzzy parameter $m$, the number of clusters $c\left(2 \leq c \leq \sqrt{n}\right)$, the maximum number of iterations $G$, the iterative convergence threshold $\varepsilon$, the missing attribute matrix $\tilde{X}_M^{(0)}$, and the membership matrix $U^{(0)}$ combined with the constraint conditions.

(2) Updating the cluster center matrix: when the algorithm performs $L$ $(L = 1, 2, \ldots)$ iterations, the cluster center $V^{(l)}$ is updated according to $U^{(l-1)}$ and (Eq. 2.3).

(3) Calculate the membership matrix: according to $V^{(l)}$, and (Eq. 2.4) solving membership matrix $U^{(l)}$.

(4) Update the missing value: calculate the missing value $\tilde{X}_M^{(l)}$ according to the membership partition matrix $U^{(l)}$ and cluster center matrix $V^{(l)}$ and (Eq. 2.18).

(5) Iteration termination: when the iteration count approaches $L = G$, or $\forall i, k$, $max\left|u_{ik}^{(l)} - u_{ik}^{(l-1)}\right| < \varepsilon$, NPS-FCM algorithm iteration stops, and the program outputs membership $U$ and cluster center $V$; or else $L = L + 1$, return (3) to continue.

# FEATURE WEIGHTED KERNEL FUNCTION FCM OF INCOMPLETE DATA

## Feature weighted FCM of incomplete data
### Feature weighted FCM algorithm with OCS

In order to solve the defects of FCM in practical application, the different contributions of FCM and sample attribute vectors to classification are considered. The sample attribute weight is introduced into the objective function, which can obtain more effective clustering analysis results. This method is called the feature weighted FCM algorithm (WFCM).

In the optimization of the complete strategy, the sample data $x_{jk}$ is composed of two segments, the complete attribute part $x_{jk}\left(o_{jk}\right)$, and the missing attribute part $x_{jk}\left(m_{jk}\right)$. Then $x_{jk}\left(o_{jk}\right) \cup x_{jk}\left(m_{jk}\right) = x_{jk}$, $x_{jk}\left(o_{jk}\right)$ remain unchanged in the clustering process. Assuming that $u_{ij}$ represents the degree of the $j$ sample data $x_j$ belonging to the $i$ cluster (the cluster center is $v_i$), $v_{ik}$ represents the $i$ feature of the $k$ cluster center, $w_{ik}$ represents the weight of the $i$ feature of the $k$ cluster center, the objective function that OCS-WFCM needs to minimize is:

$$
\begin{aligned}
&\min \sum_{i=1}^{c} \sum_{j=1}^{n} \sum_{k=1}^{l} u_{ij}^m \omega_{ik}^{\beta} \left\| x_{jk} - v_{ik} \right\|^2 \\
&s.t. \sum_{i=1}^{c} u_{ij} = 1, u_{ij} \in [0,1], \\
&\sum_{k=1}^{l} w_{ik} = 1, w_{ik} \in [0,1], \\
&i = 1, 2, \ldots, c \\
&j = 1, 2, \ldots, n \\
&k = 1, 2, \ldots, l \\
&\beta > 1
\end{aligned}
\tag{3.1}
$$

Furthermore, because of $x_{jk} = \left[x_{jk}\left(o_{jk}\right), x_{jk}\left(m_{jk}\right)\right]$, (Eq. 3.1) is equivalent to

$$
\min \sum_{i=1}^{c} \sum_{j=1}^{n} \sum_{k=1}^{l} u_{ij}^m \omega_{ik}^{\beta} \left( \left\| x_{jk}\left(o_{jk}\right) - v_{ik} \right\|^2 + \left\| x_{jk}\left(m_{jk}\right) - v_{ik} \right\|^2 \right)
\tag{3.2}
$$

Because the complete attribute $x_{jk}$ $(o_{jk})$ remains unchanged during the clustering process and is a fixed constant, the minimum value of (Eq. 3.2) can be simplified as

$$\min \sum_{i=1}^{c}\sum_{j=1}^{n}\sum_{k=1}^{l} u_{ij}^m \omega_{ik}^{\beta} \left\| x_{jk}\left(m_{jk}\right) - v_{ik} \right\|^2 \tag{3.3}$$

The optimal solution of (Eq. 3.3) can be further analyzed as

$$x_{jk}\left(m_{jk}\right) = \frac{\sum_{i=1}^{c} u_{ij}^m \omega_{ik}^{\beta} v_{ik}}{\sum_{i=1}^{c} u_{ij}^m \omega_{ik}^{\beta}} \tag{3.4}$$

In order to obtain the membership degree, cluster center and weight matrix, the Lagrange method is used to solve (Eq. 3.3).

If $x$ is known, then

$$\sum_{j=1}^{n} \lambda_j \left( \sum_{i=1}^{c} u_{ij} - 1 \right) = 0, \tag{3.5}$$

where $\lambda_j$ is the Lagrange multiplier, and $\lambda_j$ is a vector composed of the Lagrange multiplier $\lambda_1, \lambda_2, \ldots \lambda_n$.

Combining (Eq. 3.3) and (Eq. 3.5), we can get

$$\begin{aligned}
&\sum_{i=1}^{c}\sum_{j=1}^{n}\sum_{k=1}^{l} u_{ij}^m \omega_{ik}^{\beta} \left\| x_{jk}\left(m_{jk}\right) - v_{ik} \right\|^2 \\
&= \sum_{i=1}^{c}\sum_{j=1}^{n}\sum_{k=1}^{l} u_{ij}^m \omega_{ik}^{\beta} \left\| x_{jk}\left(m_{jk}\right) - v_{ik} \right\|^2 - \sum_{j=1}^{n} \lambda_j \left( \sum_{i=1}^{c} u_{ij} - 1 \right).
\end{aligned} \tag{3.6}$$

Let $Q_{ij} = \sum_{k=1}^{l} \omega_{ik}^{\beta} \left\| x_{jk}\left(m_{jk}\right) - v_{ik} \right\|^2$, further obtain

$$J_{OCS-WFCM} = \sum_{i=1}^{c}\sum_{j=1}^{n} u_{ij}^m Q_{ij} - \sum_{j=1}^{n} \lambda_j \left( \sum_{i=1}^{c} u_{ij} - 1 \right). \tag{3.7}$$

Get the partial derivative of $u_{ij}$ and get

$$\frac{\partial J_{OCS-WFCM}}{\partial u_{ij}} = m u_{ij}^{m-1} Q_{ij} - \lambda_j = 0. \tag{3.8}$$

Therefore,

$$u_{ij} = \left( \frac{\lambda_j}{m Q_{ij}} \right)^{\frac{1}{m-1}}. \tag{3.9}$$

And $\sum_{i=1}^{c} u_{ij} = 1$ is known, combined with (Eq. 3.9), we get

$$\lambda j \frac{1}{m-1} = \sum_{i=1}^{c} \left( \frac{1}{m Q_{ij}} \right)^{\frac{-1}{m-1}}. \tag{3.10}$$

Further obtained

$$u_{ij} = \frac{\sum_{r=1}^{c} \left(\frac{1}{mQ_{rj}}\right)^{\frac{-1}{m-1}}}{mQ_{ij}^{\frac{1}{m-1}}} = \left(\sum_{i=1}^{c} \frac{Q_{ij}}{Q_{rj}}\right)^{\frac{1}{1-m}}$$

$$= \left(\sum_{i=1}^{c} \frac{\sum_{k=1}^{l} \omega_{ik}^{\beta} \left\|x_{jk}\left(m_{jk}\right) - v_{ik}\right\|^2}{\sum_{k=1}^{l} \omega_{rk}^{\beta} \left\|x_{jk}\left(m_{jk}\right) - v_{ik}\right\|^2}\right)^{\frac{1}{1-m}}.$$

$$r = 1, 2, \ldots, c$$

(3.11)

Similarly, one can obtain

$$\omega_{ik} = \left(\sum_{t=1}^{l} \frac{\sum_{j=1}^{n} u_{ij}^{m} \cdot \left\|x_{jk}\left(m_{jk}\right) - v_{ik}\right\|^2}{\sum_{j=1}^{n} u_{ij}^{m} \cdot \left\|x_{jt}\left(m_{jt}\right) - v_{ik}\right\|^2}\right)^{\frac{1}{1-\beta}}.$$

(3.12)

Next, take the partial derivative of $v_{ik}$ in Eq. 3.3 to get

$$\frac{\partial J_{OCS-WFCM}}{\partial v_{ik}} = -2\sum_{j=1}^{n} u_{ij}^{m} \omega_{ik}^{\beta} \cdot \left(x_{jk}\left(m_{jk}\right) - v_{ik}\right) = 0.$$

(3.13)

Further obtained

$$v_{ik} = \frac{\sum_{j=1}^{n} u_{ij}^{m} \omega_{ik}^{\beta} x_{jk}\left(m_{jk}\right)}{\sum_{j=1}^{n} u_{ij}^{m} \omega_{ik}^{\beta}}.$$

(3.14)

It is observed by (Eq. 3.14) that when $\omega_{ik}^{\beta} = 0$, there is $v_{ik} = 0$. The formula for $v_{ik}$ is

$$v_{ik} = \begin{cases} 0, & \text{if } \omega_{ik}^{\beta} = 0 \\ \frac{\sum_{j=1}^{n} u_{ij}^{m} x_{jk}\left(m_{jk}\right)}{\sum_{j=1}^{n} u_{ij}^{m}}, & \text{if } \omega_{ik}^{\beta} \neq 0 \end{cases}$$

(3.15)

The main steps of OCS-WFCM algorithm are:

(1) Initialization: Set the fuzzy parameter $m$, number of clusters $c\left(2 \leq c \leq \sqrt{n}\right)$, utmost allowed iterations $G$, iterative convergence threshold $\varepsilon$, the missing attribute matrix $\tilde{X}_M^{(0)}$, and the membership matrix $U^{(0)}$ combined with the constraint conditions.

(2) Updating the cluster center matrix: when the algorithm performs $L$ $(L = 1, 2, \ldots)$ iterations, cluster center $V^{(l)}$ is updated according to $U^{(l-1)}$ and (Eq. 3.15).

(3) Calculate the membership matrix: according to $V^{(l)}$, and (Eq. 3.11) solving membership matrix $U^{(l)}$.

(4) Calculate the weight matrix: according to $V^{(l)}$, and (Eq. 3.12) to solve the weight matrix.

(5) Update the missing value: calculate the missing value $\tilde{X}_M^{(l)}$ according to the membership partition matrix $U^{(l)}$ and cluster center matrix $V^{(l)}$ and (Eq. 3.4).

(6) Iteration termination: when the iteration count approaches $L = G$, or $\forall i, k$, $max\left|u_{ik}^{(l)} - u_{ik}^{(l-1)}\right| < \varepsilon$, OCS-WFCM algorithm iteration stops, and the program outputs membership $U$ and cluster center $V$; or else $L = L + 1$, return (3) to continue.

*Feature weighted FCM algorithm with NPS*

In the interpolation of NPS-WFCM, the sample data $x_{jk}$ is also divided into two parts, the complete attribute part $x_{jk}\,(o_{jk})$, and the missing attribute part $x_{jk}\,(m_{jk})$. Then, $x_{jk}\,(o_{jk}) \cup x_{jk}\,(m_{jk}) = x_{jk}$, $x_{jk}\,(o_{jk})$ remain unchanged in the clustering process. The filling method of missing attributes in NPS-WFCM is as follows.

$$x_{jk}\left(o_{jk}\right) = v_{ik} = \begin{cases} 0, & if \quad \omega_{ik}^{\beta} = 0 \\ \dfrac{\sum_{j=1}^{n} u_{ij}^{m} x_{jk}}{\sum_{j=1}^{n} u_{ij}^{m}}, & if\, \omega_{ik}^{\beta} \neq 0 \end{cases}, D_{ij} = \min\left\{D_{1j}, D_{2j}, \ldots, D_{cj}\right\}. \tag{3.16}$$

Similar to OCS-WFCM, only (Eq. 3.15) needs to be replaced with (Eq. 3.16) when updating the missing attributes.

The main steps of NPS-WFCM algorithm are:

(1) Initialization: Set the fuzzy parameter $m$, number of clusters $c\,(2 \leq c \leq \sqrt{n})$, utmost allowed iterations $G$, iterative convergence threshold $\varepsilon$, the missing attribute matrix $\tilde{X}_M^{(0)}$, and the membership matrix $U^{(0)}$ combined with the constraint conditions.

(2) Updating the cluster center matrix: when the algorithm performs $L\,(L = 1, 2, \ldots)$ iterations, cluster center $V^{(l)}$ is updated according to $U^{(l-1)}$ and (Eq. 3.15).

(3) Calculate the membership matrix: according to $V^{(l)}$, and (Eq. 3.11) solving membership matrix $U^{(l)}$.

(4) Calculate the weight matrix: according to $V^l$, and (Eq. 3.12) to solve the weight matrix.

(5) Update the missing value attribute: calculate the missing value $\tilde{X}_M^{(l)}$ according to the membership partition matrix $U^{(l)}$ and cluster center matrix $V^{(l)}$ and (Eq. 3.16).

(6) Iteration termination: when the iteration count approaches $L = G$, or $\forall i, k$, $\max\left|u_{ik}^{(l)} - u_{ik}^{(l-1)}\right| < \varepsilon$, NPS-WFCM algorithm iteration stops, and the program outputs membership $U$ and cluster center $V$; or else $L = L + 1$, return (3) to continue.

## Feature weighted kernel FCM of incomplete data
### Feature weighted kernel FCM clustering with OCS

This section introduces the kernel function into the OCS-WFCM of the previous section. Clustering is performed in the kernel space, and the observed data is mapped to a higher dimensional feature space in a nonlinear way to achieve nonlinear classification techniques. It is assumed that $\varphi$ is a nonlinear mapping function, $\varphi : x \to \varphi(x) \in$ maps the high characteristic space, where $x \in X = \{x_1, x_2, \ldots, x_n\}$. $\varphi\left(x_{jk}\right)$ is the mapping of the $j$ th sample data point to the $k$ th feature in the feature space. The optimization objective function of feature weighted kernel FCM (WKFCM) with OCS is as follows.

$$\min \sum_{i=1}^{c} \sum_{j=1}^{n} \sum_{k=1}^{l} u_{ij}^{m} \omega_{ik}^{\beta} \left\| \phi\left(x_{jk}\left(m_{jk}\right)\right) - \phi\left(v_{ik}\right) \right\|^{2} \tag{3.17}$$

Expanding $\left\| \phi\left(x_{jk}\left(m_{jk}\right)\right) - \phi\left(v_{ik}\right) \right\|^{2}$ in (Eq. 3.17), we can get

$$\left\| \phi \left( x_{jk} \left( m_{jk} \right) \right) - \phi \left( v_{ik} \right) \right\|^2$$
$$= \phi \left( x_{jk} \left( m_{jk} \right) \right) \cdot \phi \left( x_{jk} \left( m_{jk} \right) \right) - 2 \phi \left( x_{jk} \left( m_{jk} \right) \right) \cdot \phi \left( v_{ik} \right) + \phi \left( v_{ik} \right) \cdot \phi \left( v_{ik} \right) \qquad (3.18)$$
$$= K \left( x_{jk} \left( m_{jk} \right), x_{jk} \left( m_{jk} \right) \right) - 2K \left( x_{jk} \left( m_{jk} \right), v_{ik} \right) + K \left( v_{ik}, v_{ik} \right)$$

Where, $K \left( x, y \right) = \phi \left( x \right) \cdot \phi \left( y \right)$ represents the kernel function, which can be used to represent the dot product in the high-dimensional feature space. The kernel function used in this work is the Gaussian kernel function, that is, $K \left( x, y \right) = \exp \left( \frac{-\| x - y \|^2}{\sigma^2} \right)$, then $K \left( x, x \right) = 1$.

Simplifying (Eq. 3.17) relative to (Eq. 3.18) yields

$$\sum_{i=1}^{c} \sum_{j=1}^{n} \sum_{k=1}^{l} u_{ij}^{m} \omega_{ik}^{\beta} \left\| \phi \left( x_{jk} \left( m_{jk} \right) \right) - \phi \left( v_{ik} \right) \right\|^2$$
$$= 2 \sum_{i=1}^{c} \sum_{j=1}^{n} \sum_{k=1}^{l} u_{ij}^{m} \omega_{ik}^{\beta} \left( 1 - K \left( x_{jk} \left( m_{jk} \right), v_{ik} \right) \right) \qquad (3.19)$$
$$= 2 \sum_{i=1}^{c} \sum_{j=1}^{n} \sum_{k=1}^{l} u_{ij}^{m} \omega_{ik}^{\beta} \left( 1 - \exp \left( \frac{-\| x_{jk} \left( m_{jk} \right) - v_{ik} \|^2}{\sigma^2} \right) \right)$$

The optimal solution of (Eq. 3.19) can be further analyzed as

$$x_{jk} \left( m_{jk} \right) = \frac{\sum_{i=1}^{c} u_{ij}^{m} \omega_{ik}^{\beta} \exp \left( \frac{-\| x_{jk} \left( m_{jk} \right) - v_{ik} \|^2}{\sigma^2} \right) v_{ik}}{\sum_{i=1}^{c} u_{ij}^{m} \omega_{ik}^{\beta} \exp \left( \frac{-\| x_{jk} \left( m_{jk} \right) - v_{ik} \|^2}{\sigma^2} \right)}. \qquad (3.20)$$

Through the Lagrange multiplier method, on the basic of on the objective function (Eq. 3.19), the updating formulas of membership degree, clustering center and weight matrix are as follows.

$$u_{ij} = \left( \sum_{i=1}^{c} \frac{\sum_{k=1}^{l} \omega_{ik}^{\beta} \left( 1 - \exp \left( \frac{-\| x_{jk} \left( m_{jk} \right) - v_{ik} \|^2}{\sigma^2} \right) \right)}{\sum_{k=1}^{l} \omega_{rk}^{\beta} \left( 1 - \exp \left( \frac{-\| x_{jk} \left( m_{jk} \right) - v_{rk} \|^2}{\sigma^2} \right) \right)} \right)^{\frac{1}{1-m}}, \qquad (3.21)$$

$$\omega_{ik}^{\beta} = \left[ \sum_{t=1}^{l} \frac{\sum_{j=1}^{n} u_{ij}^{m} \cdot \left( 1 - \exp \left( \frac{-\| x_{jk} \left( m_{jk} \right) - v_{ik} \|^2}{\sigma^2} \right) \right)}{\sum_{j=1}^{n} u_{ij}^{m} \cdot \left( 1 - \exp \left( \frac{-\| x_{jt} \left( m_{jt} \right) - v_{rt} \|^2}{\sigma^2} \right) \right)} \right]^{\frac{1}{1-\beta}}, \qquad (3.22)$$

$$v_{ik} = \begin{cases} 0, & if \ \omega_{ik}^{\beta} = 0 \\ \dfrac{\sum_{j=1}^{n} u_{ij}^{m} \exp \left( \frac{-\| x_{jk} \left( m_{jk} \right) - v_{ik} \|^2}{\sigma^2} \right) \cdot x_{jk}}{\sum_{j=1}^{n} u_{ij}^{m} \exp \left( \frac{-\| x_{jk} \left( m_{jk} \right) - v_{ik} \|^2}{\sigma^2} \right)}, & if \quad \omega_{ik}^{\beta} \neq 0 \end{cases} \qquad (3.23)$$

The main steps of OCS-WKFCM algorithm are:

(1) Initialization: Set the fuzzy parameter $m$, number of clusters $c\left(2 \leq c \leq \sqrt{n}\right)$, utmost allowed iterations $G$, iterative convergence threshold $\varepsilon$, missing attribute matrix $\tilde{X}_M^{(0)}$, membership matrix $U^{(0)}$ combined with the constraint conditions.

(2) Updating the cluster center matrix: when the algorithm performs $L$ $(L = 1, 2, \ldots)$ iterations, cluster center $V^{(l)}$ is updated according to $U^{(l-1)}$ and (Eq. 3.23).

(3) Calculate the membership matrix: according to $V^{(l)}$, and (Eq. 3.21) solving membership matrix $U^{(l)}$.

(4) Calculate the weight matrix: according to $V^{(l)}$, and (Eq. 3.22) to solve the weight matrix.

(5) Update the missing value: calculate the missing value $\tilde{X}_M^{(l)}$ according to the cluster center matrix $V^{(l)}$ and membership partition matrix $U^{(l)}$ and (Eq. 3.20).

(6) Iteration termination: when the iteration count approaches $L = G$, or $\forall i, k$, $\max\left|u_{ik}^{(l)} - u_{ik}^{(l-1)}\right| < \varepsilon$, OCS-WKFCM algorithm iteration stops, and the program outputs membership U and cluster center V; or else $L = L + 1$, return (3) to continue.

### Feature weighted kernel FCM clustering with NPS

NPS-WKFCM divides the sample data $x_{jk}$ into two parts, the complete attribute part $x_{jk}$ $(o_{jk})$ and the missing attribute part $x_{jk}$ $(m_{jk})$, then $x_{jk}\left(o_{jk}\right) \cup x_{jk}\left(m_{jk}\right) = x_{jk}$ and $x_{jk}\left(o_{jk}\right)$ remain unchanged in the clustering process. The filling method of missing attributes in NPS-WKFCM is as follows.

$$x_{jk}\left(m_{jk}\right) = v_{ik} = \begin{cases} 0, & if \quad \omega_{ik}^{\beta} = 0 \\ \dfrac{\sum_{j=1}^{n} u_{ij}^m x_{jk}}{\sum_{j=1}^{n} u_{ij}^m}, & if \, \omega_{ik}^{\beta} \neq 0 \end{cases}, \quad D_{ij} = \min\left\{D_{1j}, D_{2j}, \ldots, D_{cj}\right\}. \tag{3.24}$$

Similar to OCS-WFCM, only (Eq. 3.20) needs to be replaced with (Eq. 3.24) when updating the missing attributes.

The main steps of NPS-WFCM algorithm are:

(1) Initialization: Set the fuzzy parameter $m$, number of clusters $c\left(2 \leq c \leq \sqrt{n}\right)$, utmost allowed iterations $G$, iterative convergence threshold $\varepsilon$, missing attribute matrix $\tilde{X}_M^{(0)}$, membership matrix $U^{(0)}$ combined with the constraint conditions.

(2) Updating the cluster center matrix: when the algorithm performs $L$ $(L = 1, 2, \ldots)$ iterations, cluster center $V^{(l)}$ is updated according to $U^{(l-1)}$ and (Eq. 3.23).

(3) Calculate the membership matrix: according to $V^{(l)}$, and (Eq. 3.21) solving membership matrix $U^{(l)}$.

(4) Calculate the weight matrix: according to $V^l$, and (Eq. 3.22) to solve the weight matrix.

(5) Update the missing value: calculate the missing value $\tilde{X}_M^{(l)}$ according to cluster center matrix $V^l$ andmembership partition matrix $U^{(l)}$ and (Eq. 3.24).

(6) Iteration termination: when the iteration count approaches $L = G$, or $\forall i, k$, $\max\left|u_{ik}^{(l)} - u_{ik}^{(l-1)}\right| < \varepsilon$, NPS-WKFCM algorithm iteration stops, and the program outputs membership U and cluster center V; or else $L = L + 1$, return (3) to continue.

***The complexity of WKFCM***

An algorithm requires analysis of time complexity and space complexity. The complexity of OCS-WKFCM and NPS-WKFCM is mainly generated by clustering. In the clustering process, the number of iterations $t$, the number of clusters $c$, the dimension of sample data $l$, the number of data samples $n$ will affect the time complexity of the algorithm. Considering the worst case, the time complexity of FCM clustering algorithm is $O(Tcnl)$. In the actual calculation process, a certain amount of storage space is needed to store data needed for clustering center matrix, weight matrix, the distance between sample data points, etc. Therefore, in order to store sample data, clustering center, weight matrix, and membership matrix, the space complexity is $O(nc + nl + 2cl)$.

The storage space required for clustering centers is $O(nc)$, where nc represents the number of clustering centers. The storage space required for sample data is $O(nl)$, where nl represents the number of sample data points. The storage space required for the weight matrix and membership matrix is $O(cl)$, where cl denotes the number of relationships between the clustering centers and sample data. By summing up these storage requirements, we obtain a total space complexity of $O(nc + nl + 2cl)$.

This space complexity analysis helps us understand the storage space required during the execution of the algorithm. Through such analysis, we can evaluate the storage resource requirements of the algorithm on datasets of different scales, thus gaining a better understanding of the algorithm's utilization of space.

# EXPERIMENTAL EVALUATION

In order to verify the superiority of the proposed OCS-WFCM, NPS-WFCM, OCS-WKFCM, and NPS-WKFCM algorithms in clustering incomplete data, experiments are conducted in this section to validate them in several datasets, respectively. The dataset description and experimental steps design are described in the following subsections.

## Dataset

The UCI database is a proposed database for machine learning by the University of California Irvine (UCI) (*Bache & Lichman, 2013*). The UCI dataset is a commonly used standard test dataset. Nine real datasets are selected from them as experimental datasets, and their details are shown in Table 1.

## Experimental settings

For different datasets, the number of categories for clustering of WFCM and WKFCM models is different and needs to be determined according to the relevant attributes in different datasets. The parameters of the clustering algorithm are set uniformly. The maximum number of iterations is 200, the termination threshold is 0.0001, and the fuzzy index is 2.

To make the incomplete data generated in the experiments closer to reality, the data are processed by the random discard method, which uses different proportions set manually for the complete data to be lost randomly. Thus, an incomplete data set is generated. In this research, the missing proportions are taken as 5%, 10%, 15% and 20%. The rules for generating missing data attributes for incomplete datasets are as follows,

**Table 1  Datasets used in our experiments.**

| Dataset | Instance | Features | Classes |
|---|---|---|---|
| Iris | 150 | 4 | 3 |
| Wine | 178 | 13 | 3 |
| Breast | 277 | 9 | 2 |
| Bupa | 345 | 6 | 2 |
| Haber Man | 306 | 3 | 2 |
| Jain | 373 | 2 | 2 |
| Cmc | 1,473 | 9 | 3 |
| Waveform3 | 5,000 | 21 | 3 |
| Robotnavigation | 5,456 | 24 | 4 |

(1) In an incomplete dataset, the attribute values of sample data cannot all be lost. If the dataset is $n$-dimensional, then at most $n - 1$ attributes are lost from the incomplete data, and at least one attribute must be present in the incomplete data.

(2) In an incomplete dataset, at least one complete attribute value exists for any one-dimensional attribute, *i.e.,* the attribute column of the dataset cannot be empty to ensure the reliability of the valuation.

Each clustering algorithm performs 100 simulation experiments in each dataset with different missing proportions, and the obtained experimental results are averaged, thus reducing the chance of the experiments and the experimental errors.

## Evaluation criteria

Currently, there is no uniform evaluation index for the degree of merit of clustering algorithms. Therefore, in this work, the experimental algorithm is chosen to be evaluated from three perspectives: accuracy (Acc), iteration number, and external evaluation indexes concerning relevant literature. Among them, the external evaluation indexes are Normalized Mutual information (NMI) (*Kumar & Diwakar, 2022*), Rand Index (RI) (*Kalinichev et al., 2022*) and F$_1$-score (*Brito, Nagasharath & Wunsch, 2022*). The formulas are as follows.:

$$NMI\,(C, T) = 2\frac{\sum_{i=1}^{C}\sum_{j=1}^{T} p_{ij}\log\left(\frac{p_{ij}}{p_i \times p_j}\right)}{\sqrt{\sum_{i=1}^{C} p_i\log p_i \times \sum_{j=1}^{T} p_j\log p_j}}$$

(4.1)

$$RI = \frac{TP + TN}{N(N-1)/2}$$

(4.2)

$$F_1 = \frac{2TP}{2TP + FP + FN}.$$

(4.3)

In the formulas, matrix $G$ represents the actual classification of the samples and $T$ represents the fuzzy division of the clustering algorithm. $MI\,(G, T)$ is the mutual information of

matrices $G$ and $T$, $H(G)$ and $H(T)$ are the information entropy of matrices $G$ and $T$, respectively. The set of sample pairs in $G$ that are in the same cluster is denoted by $X$, and the set of sample pairs in $G$ that are not in the same cluster is denoted by $Z$. The fuzzy set of sample pairs in $T$ that are in the same cluster is denoted by $Y$, and the fuzzy set of sample pairs in $T$ that are not in the same cluster is denoted by $V$. Then, in the above equation, $a = |X|Y|$, $b = |X|V|$, $c = |Z|Y|$, $d = |Z|V|$.

## Experimental analysis

The missing treatment is performed on the nine datasets mentioned in Section 'Dataset', and the four optimized improvement algorithms proposed in this research are run. The results are experimentally compared with seven classical incomplete data clustering algorithms (*Shi et al., 2020*) and analyzed and described based on evaluation criteria.

To evaluate the advantages and disadvantages of the algorithms from an overall perspective, the mean values of the evaluation indexes of the 11 algorithms under the four missing ratios are taken, and the results are shown in Tables 2, 3, 4, 5 and 6.

The average ACC of the 11 algorithms with different missing rates in different datasets is reflected in Table 2. The table shows that the OCS-WFCM and NPS-WFCM algorithms proposed in this work based on feature weighting improvement have higher accuracy than the seven classical clustering algorithms under different missing rates in each dataset. The proposed OCS-WKFCM and NPS-WKFCM algorithms based on feature weighting and kernel function improvement have the highest accuracy in all datasets. The accuracy of the clustering algorithms is the most direct representation of the accuracy. This result shows that the incorporation of feature weighting and kernel methods can improve the clustering performance of the FCM algorithm for incomplete data and make it have higher clustering accuracy.

Tables 3, 4 and 5 show the calculation of three external evaluation metrics, NMI, F-score, and RI. The F-score of the proposed four optimization algorithms achieves optimality in all seven datasets except Iris and Cmc, RI achieves optimality in all eight datasets except Iris, and NMI achieves optimality in all datasets. Among them, the OCS-WFCM and NPS-WFCM algorithms are only slightly worse than the others in Bupa and Haberman datasets, and the OCS-WKFCM and NPS-WKFCM are better than the OCS-WFCM and NPS-WFCM algorithms in all datasets. Due to the random nature of missing processing, it may make too many missing features of a certain attribute, which is not conducive to updating the feature weights of OCS-WFCM and NPS-WFCM algorithms. Therefore, on the whole, the clustering accuracy of OCS-WFCM and NPS-WFCM algorithms is still better than that of the seven classical algorithms. Meanwhile, the introduction of the kernel method will alleviate the influence of feature attributes on the clustering accuracy and improve the prediction accuracy, which makes the external evaluation indexes of OCS-WFCM and NPS-WFCM algorithms better than those of OCS-WFCM and NPS-WFCM algorithms.

Table 6 shows the average number of iterations of 11 algorithms. This index mainly reflects the convergence speed of the algorithms. From the table, it can be obtained that all algorithms can reach a stable convergence state. However, in about 2/3 of the datasets, the iterations of OCS-WFCM and NPS-WFCM is significantly higher than that of the seven

**Table 2** ACC averages of different algorithms in nine datasets with different missing rates.

| Dataset Methods | ACC | | | | | | | | |
|---|---|---|---|---|---|---|---|---|---|
| | Iris | Wine | Breast | Bupa | Haber man | Jain | Cmc | Wave form3 | Robot navigation |
| ZERO | 0.574 | 0.390 | 0.517 | 0.523 | 0.498 | 0.734 | 0.358 | 0.431 | 0.458 |
| AVER | 0.789 | 0.598 | 0.538 | 0.514 | 0.519 | 0.773 | 0.385 | 0.444 | 0.499 |
| KNN | 0.813 | 0.631 | 0.567 | 0.507 | 0.540 | 0.774 | 0.402 | 0.452 | 0.494 |
| WDS | 0.830 | 0.618 | 0.580 | 0.474 | 0.510 | 0.767 | 0.375 | 0.513 | 0.491 |
| PDS | 0.832 | 0.615 | 0.592 | 0.473 | 0.514 | 0.744 | 0.390 | 0.483 | 0.472 |
| OCS | 0.827 | 0.617 | 0.596 | 0.484 | 0.532 | 0.753 | 0.385 | 0.490 | 0.480 |
| NPS | 0.832 | 0.613 | 0.606 | 0.486 | 0.548 | 0.769 | 0.393 | 0.498 | 0.487 |
| OCS-WFCM | 0.832 | 0.646 | 0.618 | 0.545 | 0.711 | 0.790 | 0.415 | 0.532 | 0.519 |
| NPS-WFCM | 0.846 | 0.648 | 0.625 | 0.549 | 0.719 | 0.792 | 0.430 | 0.541 | 0.518 |
| OCS-WKFCM | 0.851 | 0.655 | 0.635 | 0.584 | 0.750 | 0.807 | 0.469 | 0.626 | 0.529 |
| NPS-WKFCM | 0.855 | 0.656 | 0.639 | 0.587 | 0.760 | 0.807 | 0.477 | 0.625 | 0.533 |

**Table 3** NMI averages of different algorithms in nine datasets with different missing rates.

| Dataset Methods | NMI | | | | | | | | |
|---|---|---|---|---|---|---|---|---|---|
| | Iris | Wine | Breast | Bupa | Haber man | Jain | Cmc | Wave form3 | Robot navigation |
| ZERO | 0.491 | 0.337 | 0.298 | 0.250 | 0.271 | 0.226 | 0.367 | 0.303 | 0.169 |
| AVER | 0.662 | 0.369 | 0.362 | 0.234 | 0.311 | 0.314 | 0.431 | 0.308 | 0.190 |
| KNN | 0.709 | 0.426 | 0.368 | 0.221 | 0.346 | 0.326 | 0.462 | 0.311 | 0.180 |
| WDS | 0.752 | 0.384 | 0.388 | 0.154 | 0.294 | 0.318 | 0.399 | 0.317 | 0.179 |
| PDS | 0.753 | 0.383 | 0.385 | 0.148 | 0.298 | 0.246 | 0.437 | 0.314 | 0.172 |
| OCS | 0.749 | 0.379 | 0.385 | 0.173 | 0.340 | 0.308 | 0.423 | 0.314 | 0.174 |
| NPS | 0.753 | 0.380 | 0.386 | 0.177 | 0.359 | 0.300 | 0.442 | 0.315 | 0.174 |
| OCS-WFCM | 0.761 | 0.458 | 0.410 | 0.336 | 0.513 | 0.344 | 0.495 | 0.320 | 0.228 |
| NPS-WFCM | 0.766 | 0.461 | 0.411 | 0.343 | 0.521 | 0.346 | 0.510 | 0.324 | 0.234 |
| OCS-WKFCM | 0.777 | 0.502 | 0.414 | 0.424 | 0.571 | 0.372 | 0.559 | 0.337 | 0.242 |
| NPS-WKFCM | 0.775 | 0.502 | 0.417 | 0.430 | 0.594 | 0.368 | 0.567 | 0.341 | 0.245 |

classical algorithms. In all the datasets, the iterations of OCS-WKFCM and NPS-WKFCM are lower than that of OCS-WFCM and NPS-WFCM. The results show that the feature weights will increase the number of iterations in some datasets, while the kernel method will significantly reduce the number of iterations. The kernel method will significantly reduce the number of iterations and improve the solving speed of the algorithm.

Compared with AVER-FCM, ZERO-FCM, and KNN-FCM algorithms, the four algorithms proposed in this research are superior. AVER-FCM, ZERO-FCM, and KNN-FCM fill the missing attributes with 0 values, sample mean values, and mean values of K neighboring samples, respectively, and then run the FCM algorithm. 0-value interpolation and mean interpolation will make the samples lose a large amount of data information,

**Table 4  F-score averages of different algorithms in nine datasets with different missing rates.**

| Dataset Methods | F-score | | | | | | | | |
|---|---|---|---|---|---|---|---|---|---|
| | Iris | Wine | Breast | Bupa | Haber man | Jain | Cmc | Wave form3 | Robot navigation |
| ZERO | 0.740 | 0.528 | 0.603 | 0.583 | 0.536 | 0.750 | 0.305 | 0.509 | 0.462 |
| AVER | 0.858 | 0.630 | 0.610 | 0.573 | 0.574 | 0.787 | 0.346 | 0.539 | 0.493 |
| KNN | 0.873 | 0.693 | 0.612 | 0.562 | 0.615 | 0.800 | 0.395 | 0.546 | 0.491 |
| WDS | 0.888 | 0.648 | 0.616 | 0.506 | 0.555 | 0.783 | 0.330 | 0.567 | 0.491 |
| PDS | 0.886 | 0.649 | 0.618 | 0.503 | 0.558 | 0.760 | 0.351 | 0.550 | 0.485 |
| OCS | 0.886 | 0.661 | 0.618 | 0.524 | 0.603 | 0.783 | 0.348 | 0.552 | 0.488 |
| NPS | 0.890 | 0.653 | 0.619 | 0.528 | 0.624 | 0.768 | 0.357 | 0.557 | 0.489 |
| OCS-WFCM | 0.885 | 0.716 | 0.623 | 0.630 | 0.769 | 0.853 | 0.391 | 0.610 | 0.509 |
| NPS-WFCM | 0.897 | 0.718 | 0.624 | 0.637 | 0.781 | 0.851 | 0.408 | 0.624 | 0.514 |
| OCS-WKFCM | 0.893 | 0.763 | 0.630 | 0.668 | 0.822 | 0.854 | 0.463 | 0.636 | 0.519 |
| NPS-WKFCM | 0.903 | 0.764 | 0.632 | 0.677 | 0.817 | 0.860 | 0.472 | 0.638 | 0.521 |

**Table 5  RI averages of different algorithms in nine datasets with different missing rates.**

| Dataset Methods | RI | | | | | | | | |
|---|---|---|---|---|---|---|---|---|---|
| | Iris | Wine | Breast | Bupa | Haber man | Jain | Cmc | Wave form3 | Robot navigation |
| ZERO | 0.737 | 0.589 | 0.659 | 0.428 | 0.371 | 0.609 | 0.390 | 0.640 | 0.525 |
| AVER | 0.843 | 0.662 | 0.666 | 0.413 | 0.410 | 0.648 | 0.434 | 0.647 | 0.559 |
| KNN | 0.862 | 0.703 | 0.667 | 0.416 | 0.523 | 0.654 | 0.456 | 0.652 | 0.542 |
| WDS | 0.873 | 0.677 | 0.673 | 0.355 | 0.394 | 0.644 | 0.413 | 0.658 | 0.536 |
| PDS | 0.875 | 0.672 | 0.675 | 0.346 | 0.398 | 0.620 | 0.438 | 0.654 | 0.529 |
| OCS | 0.873 | 0.689 | 0.676 | 0.381 | 0.430 | 0.644 | 0.431 | 0.657 | 0.531 |
| NPS | 0.880 | 0.677 | 0.680 | 0.389 | 0.454 | 0.627 | 0.444 | 0.658 | 0.533 |
| OCS-WFCM | 0.878 | 0.727 | 0.699 | 0.469 | 0.571 | 0.676 | 0.483 | 0.662 | 0.566 |
| NPS-WFCM | 0.885 | 0.729 | 0.700 | 0.475 | 0.581 | 0.674 | 0.501 | 0.667 | 0.569 |
| OCS-WKFCM | 0.883 | 0.737 | 0.714 | 0.526 | 0.638 | 0.693 | 0.551 | 0.676 | 0.589 |
| NPS-WKFCM | 0.890 | 0.741 | 0.716 | 0.525 | 0.656 | 0.695 | 0.564 | 0.678 | 0.591 |

which is the most basic interpolation strategy. The KNN algorithm is extremely data-dependent, and individual data anomalies will affect the effect of the whole clustering. The traversal mechanism of the KNN algorithm is prone to dimensional disasters on large datasets. At the same time, the above algorithms fill in the missing data in the sample and then perform clustering. The data filling algorithm will have certain errors in filling accuracy and cannot accurately represent the missing data, and then clustering on the filled data set will have even lower clustering accuracy. The four improved algorithms are based on OCS-FCM and NPS-FCM algorithms, which dynamically update the incomplete data during the clustering iterations and organically combine clustering and interpolation. This

**Table 6  Iterations averages of different algorithms in nine datasets with different missing rates.**

| Dataset Methods | Iterations | | | | | | | | |
|---|---|---|---|---|---|---|---|---|---|
| | Iris | Wine | Breast | Bupa | Haber man | Jain | Cmc | Wave form3 | Robot navigation |
| ZERO | 60.20 | 68.90 | 27.94 | 51.45 | 57.48 | 31.47 | 37.49 | 28.96 | 27.97 |
| AVER | 33.05 | 43.02 | 28.67 | 36.38 | 24.41 | 25.24 | 24.54 | 27.15 | 27.06 |
| KNN | 41.33 | 43.70 | 28.63 | 36.64 | 37.79 | 29.54 | 35.51 | 25.37 | 29.24 |
| WDS | 26.40 | 47.39 | 33.13 | 37.48 | 25.65 | 19.72 | 23.48 | 26.99 | 25.88 |
| PDS | 26.75 | 41.49 | 28.04 | 38.30 | 26.24 | 29.94 | 24.40 | 31.40 | 24.79 |
| OCS | 33.63 | 55.56 | 25.09 | 43.85 | 27.20 | 29.14 | 26.38 | 34.98 | 27.37 |
| NPS | 28.75 | 52.67 | 27.32 | 39.65 | 26.66 | 27.98 | 25.74 | 35.63 | 25.37 |
| OCS-WFCM | 36.23 | 46.18 | 30.90 | 35.43 | 37.34 | 22.05 | 28.05 | 30.37 | 30.69 |
| NPS-WFCM | 31.20 | 42.75 | 29.10 | 34.35 | 36.93 | 19.02 | 27.33 | 26.61 | 28.00 |
| OCS-WKFCM | 34.85 | 42.30 | 29.23 | 34.00 | 30.64 | 18.94 | 26.85 | 28.59 | 27.05 |
| NPS-WKFCM | 29.25 | 41.13 | 27.08 | 32.25 | 26.99 | 17.74 | 25.31 | 24.49 | 24.45 |

avoids the secondary accuracy reduction caused by the algorithms to some extent and has better robustness.

Compared with the WDS-FCM, PDS-FCM, OCS-FCM, and NPS-FCM algorithms, the OCS-WFCM and NPS-WFCM algorithms are superior. The WDS-FCM algorithm discards incomplete data samples, which will have a greater impact on the clustering results in the case of high missing data samples and reduce the overall sample size. The PDS-FCM algorithm is an improvement of the WDS-FCM algorithm but does not deal with missing attributes. Both algorithms do not treat missing attributes, and the data information is wasted. Its information value is not maximized, and the clustering results are unsatisfactory. The traditional OCS-FCM and NPS-FCM do not consider the role played by different features in the clustering process and treat all features equally. In contrast, the OCS-WFCM and NPS-WFCM algorithms assign weights to different features on this basis. At the same time, dynamic adjustments are made during the iterative process to minimize the influence of outlier points in the sample on the clustering center. This results in a better clustering effect in most of the datasets.

Based on the OCS-WFCM and NPS-WFCM algorithms, a greater improvement is made in this work. The OCS-WKFCM and NPS-WKFCM algorithms are proposed. The above modification introduces the kernel method into the FCM algorithm for incomplete data and solves the nonlinear separable problem between clusters and clusters in complex data. The number of iterations of the algorithm is substantially reduced based on the improved clustering, which makes the algorithm perform better.

Figures 1, 2, 3 and 4 show the specific performance of the evaluation criteria, ACC, NMI, F-score, and RI, respectively, in different datasets and missing proportions. Among them, ZERO-FCM, AVER-FCM, KNN-FCM, WDS-FCM, and PDS-FCM only have good accuracy in partial datasets and fluctuate greatly in some missing proportions. Compared with the above five algorithms, OCS-FCM and NPS-FCM algorithms are not optimal

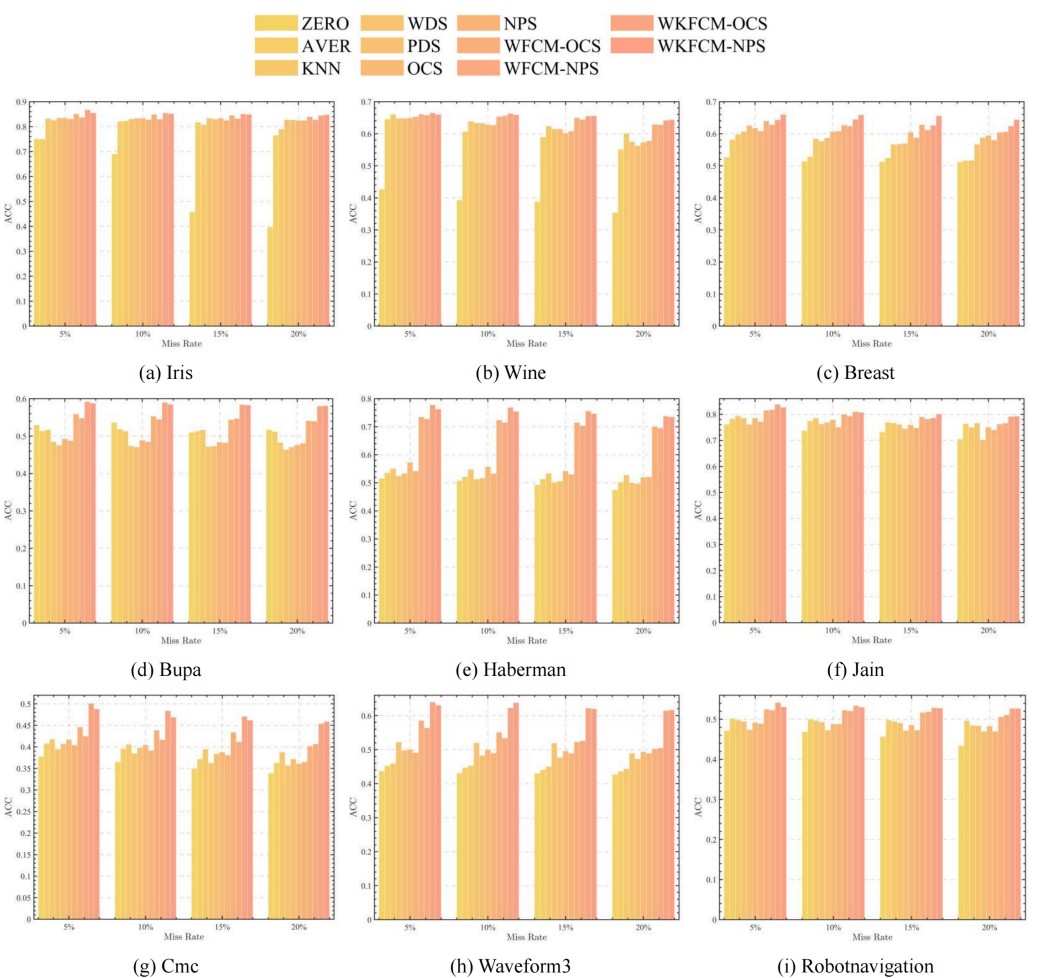

**Figure 1** Histogram of ACC averages in nine datasets with different missing values.

in all cases, but the clustering accuracy starts to maintain stability. Compared with the OCS-FCM and NPS-FCM algorithms, the proposed four algorithms all showed significant improvement in clustering accuracy. This indicates that the optimization algorithms continue the advantages of the original algorithms and still have better robustness. Meanwhile, the histogram distribution in the figure shows that the OCS-WKFCM algorithm possesses higher evaluation criteria values and better clustering accuracy for low missing rates of only 5%–10%. The NPS-WKFCM algorithm provides higher accuracy for high missing rates of 15–20%.

Considered from the perspective of interpolation methods, the OCS-WKFCM algorithm takes into account the information of missing data attributes. It can still maintain the excellent performance of the FCM algorithm as the missing rate increases and keep the clustering accuracy stable. However, the OCS-WKFCM algorithm requires repeated iterations to update the missing attribute values, which can make the number of iterations of the algorithm increase significantly. The NPS-WKFCM algorithm updates the missing

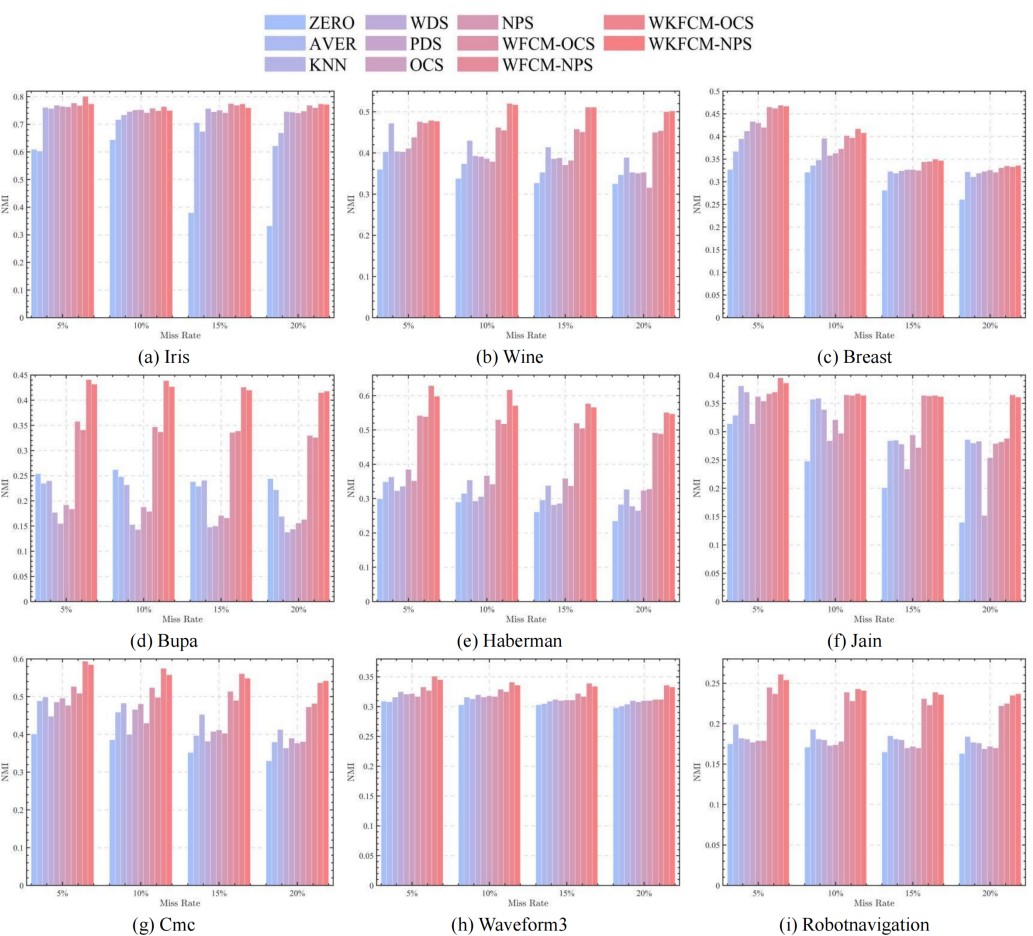

**Figure 2** Histogram of NMI averages in nine datasets with different missing values.

values by comparing them with the clustering centers derived from the current iteration. It no longer requires repeated iterations and reduces the difficulty of solving. The experimental comparison reveals that its accuracy is better with a high missing rate.

## CONCLUSION

For incomplete data clustering, a new generalized fuzzy clustering framework incorporating feature weights and kernel methods is developed in this work. The four improved algorithms specifically involved are WFCM-OCS, WFCM-NPS, WKFCM-OCS, and WKFCM-NPS. The experimental results validate the effectiveness of the proposed framework and show that the optimized algorithms are superior in the clustering of incomplete data. Meanwhile, the following conclusions are drawn:

(1) The improvement based on feature weights can improve the clustering precision of the FCM algorithm in most incomplete datasets. However, it also dramatically raises the iteration number and increase the complexity of the algorithm.

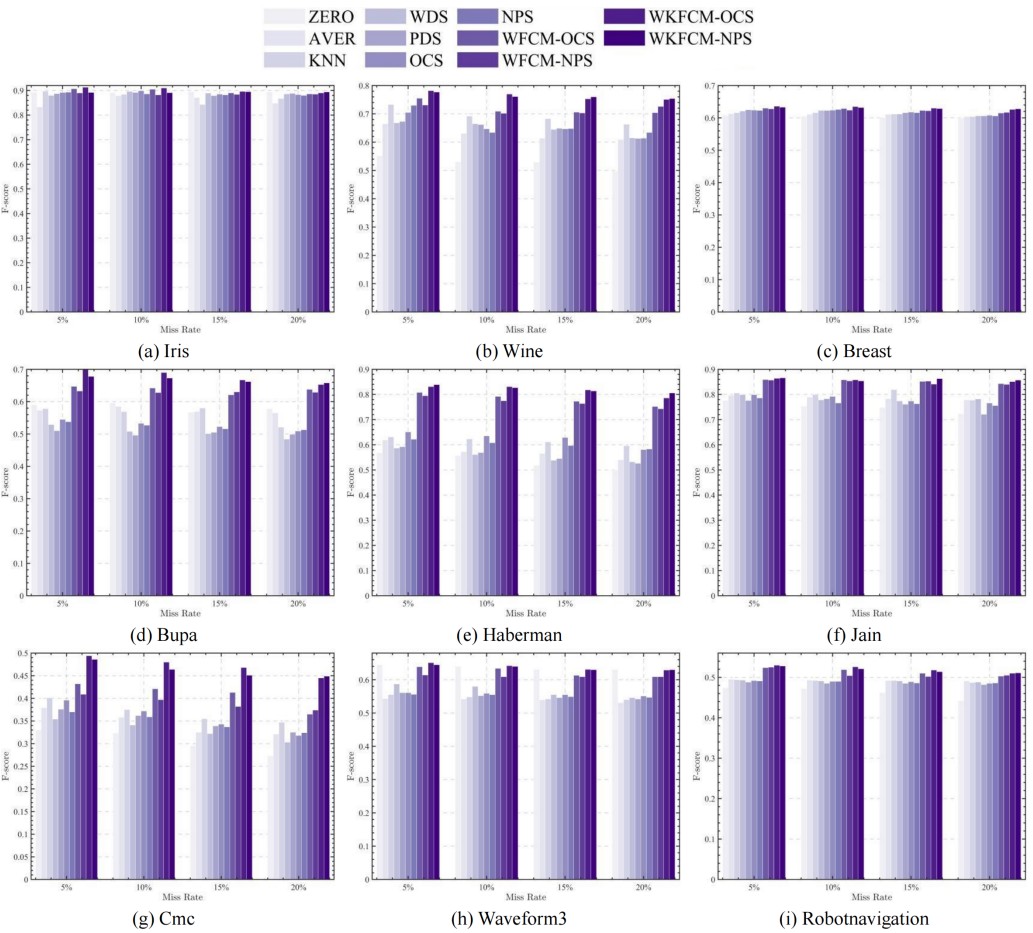

**Figure 3** Histogram of F-score averages in nine datasets with different missing values.

(2) On the basis of the OCS-WFCM and NPS-WFCM algorithms, the data are mapped by the kernel method for high latitude mapping can effectively improve the clustering accuracy, and does not influence iteration number significantly.

(3) The OCS-WKFCM algorithm has higher clustering precision at low missing rate of 5%–10%, while the NPS-WKFCM performs better at high missing rate of 15–20%.

The mathematical model in this study has significant generalization ability and interpretability, which are crucial for practical applications. The model performs consistently and reliably on different datasets and adapts to multiple domains and application scenarios. Meanwhile, the interpretability of the model enables decision makers to understand and accept the results of the model. These properties drive the success of real-world applications.

The OCS-WKFCM algorithm and NPS-WKFCM algorithm proposed in this paper combine two different techniques and provide new ideas and methods for data analysis and processing. They can effectively regulate the distribution of feature weights through fuzzy indices, and can significantly reduce the impact of feature weighting on performance

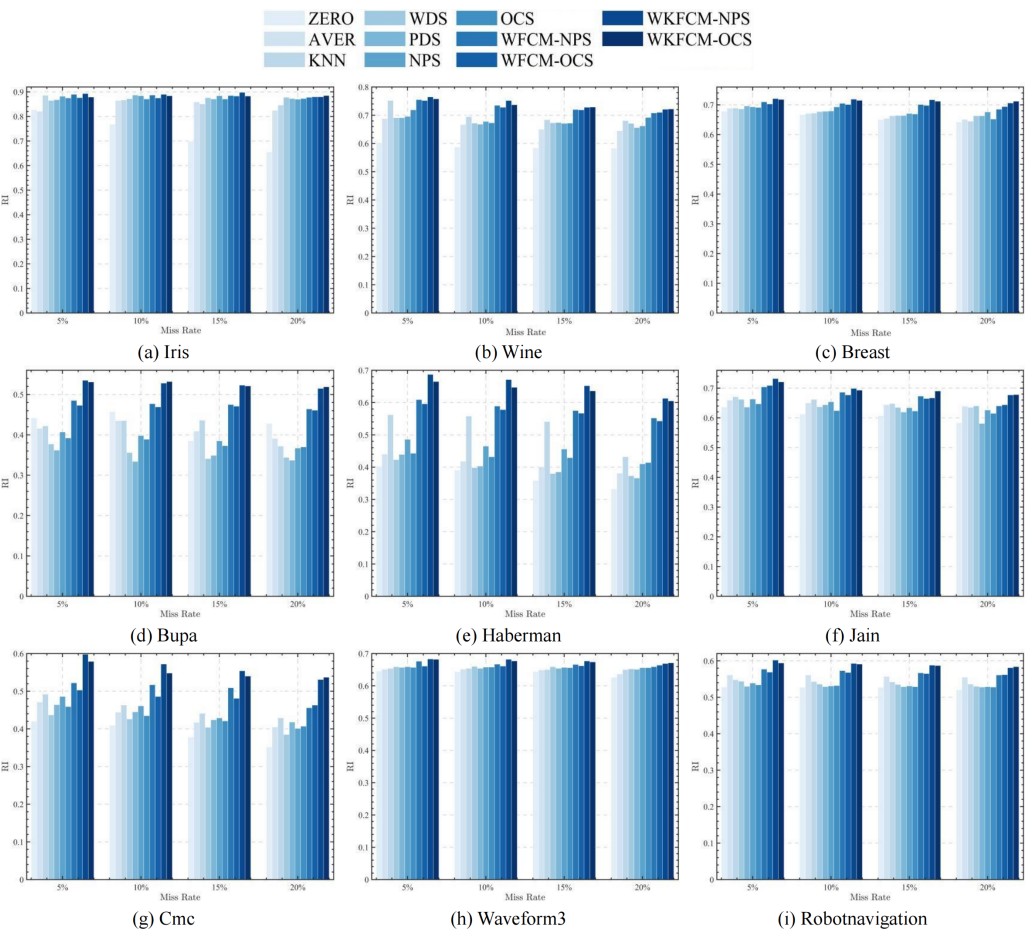

**Figure 4  Histogram of RI averages in nine datasets with different missing rates.**

when the product approach is used. However, the performance of the model is highly dependent on the data quality and features.

Further research can be pursued in the following directions:

(1) Further improvement of the optimization strategies for NPS and OCS to effectively handle fuzziness and strike a balance in optimizing clustering results.

(2) Exploration of the possibilities of integrating these strategies with other clustering algorithms and data processing techniques, such as feature selection and dimensionality reduction, to enhance cluster performance and efficiency.

(3) Dedicated efforts to advance the field of clustering, addressing practical challenges encountered in real-world applications, and enhancing the accuracy and reliability of clustering results.

### Funding

The authors received no funding for this work.

### Competing Interests

The authors declare there are no competing interests.

### Author Contributions

- Ying Yang conceived and designed the experiments, performed the experiments, performed the computation work, authored or reviewed drafts of the article, and approved the final draft.
- Haoyu Chen conceived and designed the experiments, analyzed the data, performed the computation work, prepared figures and/or tables, authored or reviewed drafts of the article, and approved the final draft.
- Haoshen Wu conceived and designed the experiments, performed the experiments, analyzed the data, prepared figures and/or tables, and approved the final draft.

### Data Availability

The data is available at the following:

- Iiris: https://archive-beta.ics.uci.edu/dataset/53/iris
- Wine: https://archive-beta.ics.uci.edu/dataset/109/wine
- Haberman: https://archive-beta.ics.uci.edu/dataset/43/haberman+s+survival
- Robotnavigation: https://archive-beta.ics.uci.edu/dataset/194/wall+following+robot+navigation+data
- Waveform3: https://archive-beta.ics.uci.edu/dataset/107/waveform+database+generator+version+1
- Breast: https://archive-beta.ics.uci.edu/dataset/15/breast+cancer+wisconsin+original

The raw data is available at the UC Irvine Machine Learning Repository: https://archive-beta.ics.uci.edu/.

### Supplemental Information

Supplemental information for this article can be found online at http://dx.doi.org/10.7717/peerj-cs.1600#supplemental-information.

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
