# Peer review of "A generalized fuzzy clustering framework for incomplete data by integrating feature weighted and kernel learning"

_PeerJ Computer Science, doi:10.7717/peerj-cs.1600_

## Round 0.1 · original submission · Major Revisions

Based on reviewers' comments the manuscript needs revisions.

In particular, Ref#3 comments on the content overlap MUST be responded to.

Reviewer 1 ·

Basic reporting

1- There are several grammatical errors in the text that cause some ambiguity.
2- The referenced literature is relevant but there are several incompatible citations for instance on line 46 and line 59. The citation form should be related to the outline of the journal and should be kept consistent during the text.
3- While the raw data have been shared and can be accessed publicly, in terms of structure, both table and figures need modifications. For instance, in the tables, name of the methods appear differently from whatever used in the text. (in text: OCS-WFCM and in tables: WFCM-OCS). The figures are too small for all the details to be visually graspable. In all the figures:
a. What does color shades mean?
b. If the tones are the same in all if nine charts then why each have a tiny legend from which the reader cannot understand anything? Remove small legend and add a large legend on top of the 9 charts (1 legend for all nine charts).
4- The manuscript is self-contained, all the results are relevant and complete.
5- Terms and theorems have been defined and proven adequately.

Experimental design

1- Proposing methods to enhance the existing body of knowledge in the field of clustering data when there are also some missing data, this research is original and within the scope of this journal.
2- The goal of the manuscript has been clearly explained and the research questions have been answered in the text.
3- Proofs and methods are sufficiently described.

Validity of the findings

1- The results’ validation has been done via the application of the proposed methods on different benchmark data sets. The proposed methods perform better than other existing clustering methods in case of having some missing data. The authors considered and explained different measures to do the comparisons of performance among all methods.
2- Results, underlying data and calculation formula are all explained which makes the replication of the results easy and straightforward.
3- Conclusions are written based on the research question.

Additional comments

For all the other comments please check the attached file which is my handwritten notes on the print out of the manuscript. All the issues should be addressed carefully and thoroughly. The notes are mainly about grammatical errors and English language usage in the way that cause ambiguity.

Annotated reviews are not available for download in order to protect the identity of reviewers who chose to remain anonymous.

Reviewer 2 ·

Basic reporting

In this paper, a generalized fuzzy clustering framework is proposed based on OCS and NPS for incomplete data. Specifically, four incomplete data fuzzy clustering algorithms are developed, and considers the contributions of different features to clustering. The authors have shown the rationality and effectiveness of the proposed method through many real datasets. From a general perspective, I think this work is useful and meaningful, as it owns many excellent properties and is useful for incomplete data. With the aim of recommending for publication, some suggestions are presented for you to improve the paper.

-The tense in the abstract and throughout the article should be consistent, and it is recommended to use the present tense.
-‘OCS’ and ‘NPS’ are the core of this article and should be used as keywords.
-Line 31: "Clustering analysis[1] is an unsupervised learning technique, which can autonomously classify data without a priori knowledge." - 'a priori' should be italicized, as it is a Latin phrase used in English.
-Line 39: The hyphen in "real - world" should not be surrounded by spaces. It should be "real-world".
-Line 71: "Doquire and Veleysen[14] estimated the missing values of fragmentary data using a KNN method based on mutual information.Tutz and Ramzan[15] proposed" - There should be a space after the period between 'information.' and 'Tutz'.
-Line 119: "we unify imputation, feature learning and" - A comma is needed after "learning" for correct use of the Oxford comma.
-The authors reviewed a lot of work based on OCS and NPS. What are the shortcomings of these works?
- In introduction, there is a lack of review and analysis of recent relevant literature.
- Discuss the existing research gaps and how the current research fills these gaps.
- In Section 3, the expression of the objective function should be unified with the previous one, for example, (3.1) does not reflect the variables to be optimized.
- What are the limitations of the proposed mathematical model?
- The value of the parameter \beta does not seem to be given.
- The evaluation indicators used to add relevant literature.
- Please explain with better technical depth, such as generalizability analysis.
- Please explain with better technical depth on originality and advantages.
- The description of future work is too rough.
- Add some recent papers in this field in references. For example:
doi.org/10.22105/jarie.2021.276107.1270 (https://doi.org/10.22105/jarie.2021.276107.1270) doi.org/10.22105/bdcv.2022.332454.1061 (https://doi.org/10.22105/bdcv.2022.332454.1061)
doi.org/10.22105/riej.2021.291738.1229 (https://doi.org/10.22105/riej.2021.291738.1229)
doi.org/10.22105/jarie.2021.307635.1387 (https://doi.org/10.22105/jarie.2021.307635.1387)

Experimental design

no comment

Validity of the findings

no comment

Additional comments

no comment

Reviewer 3 ·

Basic reporting

The paper is excessively non connected and certain sections are unclear. The writing in some parts lacks directness and employs overly complex language. It would be beneficial to adopt a more straightforward approach.

I observe a bias in the choice of cited articles and state-of-art review, some papers cited without access or in terms of prepublish that do not allow evaluation of originality of the paper.

The notation and graphics are of low quality, and the document presents a successive number of typos.

There is no clear evidence that in fact the proposed algorithms use stopping criteria and algorithmic convergence in order to guarantee or control classification errors.

Experimental design

I'm not particularly enthusiastic about the proposed approach itself. The authors propose a generalized fuzzy clustering framework for incomplete data by integrating feature weighted and kernel learning but this is a very limited contribution to science since other proposals like the ones given in [1] have already been proposed in the literature and I find a lot of similarity between the proposed work and this paper.

[1] Rodrigues, A. K., Ospina, R., & Ferreira, M. R. (2021). Adaptive kernel fuzzy clustering for missing data. Plos one, 16(11), e0259266.

- Only the NPS proposal seems to be a methodological contribution but it lacks a solid discussion that deserves a publication

Validity of the findings

- The rationality of the experimental studies is not clear and requires further explanations and details. The experimental study setting is highly controlled and does not present significant challenges to the procedure or the clustering once since we are only looking at classification and not at the quality of the algorithm dealing with missing data, which can produce biased results.

- In order to properly evaluate the proposed approach, it is necessary to include scenarios involving contamination and clearly specification on missing generator mechanism. These scenarios will aid in assessing the effectiveness of the proposed method under more realistic and challenging conditions.

Additional comments

No comments

---

## Round 0.2 · accepted · Accept

The Authors have revised the manuscript thoroughly and carefully based on all the comments. I accept this article.

Reviewer 1 ·

Basic reporting

All the major and minor concerns, clarification, corrections, and typos that I asked to be considered in the revised version have been addressed by the authors. I suggest the publication of the paper as is.

Experimental design

All the major and minor concerns, clarification, corrections, and typos that I asked to be considered in the revised version have been addressed by the authors. I suggest the publication of the paper as is.

Validity of the findings

All the major and minor concerns, clarification, corrections, and typos that I asked to be considered in the revised version have been addressed by the authors. I suggest the publication of the paper as is.

Additional comments

All the major and minor concerns, clarification, corrections, and typos that I asked to be considered in the revised version have been addressed by the authors. I suggest the publication of the paper as is.

Reviewer 2 ·

Basic reporting

Upon reviewing the updated manuscript, I find that the authors have adequately addressed my previous concerns and incorporated the necessary changes. The revisions have significantly improved the clarity and quality of the paper. I believe the manuscript, in its current form, is suitable for publication and meets the standards of PeerJ Computer Science.

Experimental design

NA

Validity of the findings

NA

Additional comments

NA

·

Basic reporting

Ok.

Experimental design

Ok.

Validity of the findings

Ok.

Additional comments

Ok.